# Metabolic Reprogramming of Ovarian Cancer Spheroids during Adhesion

**DOI:** 10.3390/cancers14061399

**Published:** 2022-03-09

**Authors:** Stephanie L. E. Compton, Joseph P. Grieco, Benita Gollamudi, Eric Bae, Jennifer H. Van Mullekom, Eva M. Schmelz

**Affiliations:** 1Department of Human Nutrition, Foods and Exercise, Virginia Tech, Blacksburg, VA 24060, USA; sedwrds3@vt.edu; 2TBMH, Fralin Biomedical Research Institute Virginia Tech, Roanoke, VA 24060, USA; gjoe2@vt.edu; 3School of Neuroscience, Virginia Tech, Blacksburg, VA 24061, USA; benitag@vt.edu; 4Statistical Applications and Innovations Group, Department of Statistics, Virginia Tech, Blacksburg, VA 24061, USA; ericbae@vt.edu (E.B.); vanmuljh@vt.edu (J.H.V.M.)

**Keywords:** ovarian cancer, metabolism, respiration, glucose uptake, glutamine, spheroid, hypoxia, substrate utilization, mitochondrial function, sphingosine-1-phosphate

## Abstract

**Simple Summary:**

During metastasis, ovarian cancer spheroids move from the original tumor and travel through the abdominal cavity to other sites. During the movement of spheroids, cells are exposed to an environment with little oxygen or glucose, unlike how cells are traditionally cultured. In this study, we used culture conditions relevant to the abdominal cavity to investigate how energy production and fuel sources change as spheroids adhere to a secondary location. Using early time points of adhesion, we show that over time, spheroids change their metabolism as well as fuel sources as they grow out. This suggests that adhesion to another site initiates a metabolic switch that supports outgrowth and successful metastasis. This is important because if we understand how these cancer cells shift from a dormant into a growing state we can identify drug targets that may suppress these signaling pathways leading to the metabolic switch and preventing the ovarian metastases from growing and invading other tissues. This may prolong the life of women with ovarian cancer.

**Abstract:**

Ovarian cancer remains a deadly disease and its recurrence disease is due in part to the presence of disseminating ovarian cancer aggregates not removed by debulking surgery. During dissemination in a dynamic ascitic environment, the spheroid cells’ metabolism is characterized by low respiration and fragmented mitochondria, a metabolic phenotype that may not support secondary outgrowth after adhesion. Here, we investigated how adhesion affects cellular respiration and substrate utilization of spheroids mimicking early stages of secondary metastasis. Using different glucose and oxygen levels, we investigated cellular metabolism at early time points of adherence (24 h and less) comparing slow and fast-developing disease models. We found that adhesion over time showed changes in cellular energy metabolism and substrate utilization, with a switch in the utilization of mostly glutamine to glucose but no changes in fatty acid oxidation. Interestingly, low glucose levels had less of an impact on cellular metabolism than hypoxia. A resilience to culture conditions and the capacity to utilize a broader spectrum of substrates more efficiently distinguished the highly aggressive cells from the cells representing slow-developing disease, suggesting a flexible metabolism contributes to the stem-like properties. These results indicate that adhesion to secondary sites initiates a metabolic switch in the oxidation of substrates that could support outgrowth and successful metastasis.

## 1. Introduction

Ovarian cancer is the deadliest gynecological cancer and remains the fifth leading cause of cancer death in women [1] due in part to a late diagnosis after metastasis has occurred [2]. During disease progression, cells or cell clusters are exfoliated from the original tumor on the ovaries or fallopian tubes [3,4] and aggregate to form multicellular spheroids [5]. The spheroids are then spread throughout the peritoneal cavity by the flow of peritoneal fluid or the malignant ascites at later stages, allowing attachment to secondary sites and initiation of metastasis. Aggregation has been shown to provide the cancer cells signals that improve survival [6,7], resistance to drug therapies [8], and has been associated with an invasive phenotype [9,10]. The presence of cancer spheroids during disease progression is unlikely to be completely removed through traditional debulking surgery [2], thus contributing to higher rates of disease recurrence and cancer death.

Advanced stages of disease and metastasis are often associated with an increased volume of ascites within the peritoneal cavity, due to leaky vasculature in the peritoneal wall or tumors releasing higher volumes of fluid, and obstruction of the lymph vessels that reduce drainage of the peritoneal fluid [11]. Ascites is associated with poorer outcomes and lower survival of ovarian cancer patients [11,12]. During dissemination and within the movement through ascites, ovarian metastases are unvascularized [5] and exposed to an oxygen content that can be as low as <1.5% [13,14]. Diffusion of oxygen from the environment into spheroids is dependent upon size, meaning that larger spheroids may contain a hypoxic core [15]. While very little information is available on nutrient levels in the malignant ascites, glucose levels have also been shown to be significantly lower in patients with malignant ascites [16,17].

In addition to cyto- and chemokines released from recruited immune cells, another characteristic of interest of the malignant ascites in ovarian cancer patients is the presence of bioactive lipid molecules, particularly sphingosine-1-phosphate (S1P). S1P has been found to be significantly increased in blood and tumor tissue of ovarian cancer patients [18,19,20] and ascites [21]. Elevated levels of S1P denote a poor prognosis and have been suggested as a possible biomarker for ovarian cancer [22]. S1P regulates migration, proliferation, and invasion of cancer cells [23] and the production of S1P is an activator of angiogenesis [24,25,26]. S1P production can be enhanced by hypoxia [24,26]. Taken together, the ascitic microenvironment that ovarian spheroids are exposed to is hypoxic with low access to glucose but elevated levels of bioactive molecules. These factors could contribute to the survival and a more aggressive phenotype of the disseminating spheroids.

Accelerated glycolysis and lactate production are well-known characteristics of tumor cells known as the Warburg effect [27,28]. Accordingly, previous studies from our lab have shown that adherent cells become more glycolytic during progression in normoxic conditions [8]. We have also shown that highly aggressive cells show greater metabolic flexibility [29,30] that can contribute to their stem-like phenotype. Warburg suggested that dysfunctional mitochondria are the cause of the cancer cells’ glycolytic metabolism. However, while the metabolic changes in ovarian cancer cells were associated with an altered mitochondrial organization in adherent cells and fragmentation in spheroids, in both conditions the mitochondria were fully functional [31]. Most strikingly, aggregation caused a drastic downregulation of metabolism, particularly in hypoxia [8]. Changes in cellular metabolism and TCA cycle enzymes as a response to aggregation have been described before [32] albeit not in hypoxic conditions and have been proposed to be the first metastatic step in exfoliated cells [33]. Our studies are in agreement with the current understanding of mitochondrial function in cancer that state the “selfish” reprograming of metabolism of the cancer cells and cells in their microenvironment is critical to enhancing the survival of the cancer cells (see recent review [34]). Thus, the modification of cellular metabolism as a response to the microenvironmental context and cues is vital to support energetic/anabolic needs and metabolic flexibility. This flexibility, rather than increased glycolysis, should be a hallmark of successful cancer cells. Consequently, we hypothesized that the adherence of ovarian cancer spheroids will increase the low respiration and ATP synthesis that support viability in limiting conditions during dissemination to allow for secondary outgrowth. In the present study, we explored metabolic changes and adaptations that occur upon adherence of ovarian cancer spheroids, mimicking the first steps towards metastasis at distant sites. Using early points of adherence (24 h and less), we found a switch from glutamine to glucose use, a differential use of substrates between cells representing slow- and fast-developing disease, and a resilience to low oxygen and glucose levels that were associated with an aggressive cancer phenotype. Identifying these early metabolic switches upon secondary adhesion will allow for a more complete understanding of the metabolic changes that happen throughout the metastatic cycle of ovarian cancer, as well as identify potential druggable metabolic targets to prevent secondary outgrowth and, thus, metastasis.

## 2. Materials and Methods

### 2.1. Cell Culture

A syngeneic murine cell model for progressive ovarian cancer was previously developed as described [35]. These cell lines express fallopian tube markers [36] and are therefore likely representing serous ovarian cancer. A late-stage mouse ovarian surface epithelial (MOSE-L) line was used to generate the more aggressive tumor-initiating cell variant (MOSE-L_TIC*v*_) which displays increased tumorigenicity [37,38,39]. Both MOSE-L and MOSE-L_TIC*v*_ form viable spheroids. MOSE cells were routinely cultured in high glucose Dulbecco’s Modified Eagle’s Medium (DMEM, Sigma-Aldrich, St. Louis, MO, USA) supplemented with 4% fetal bovine serum (Atlanta Biological, Flowery Branch, GA, USA) and 100 µg/mL of each penicillin and streptomycin at 37 °C in a humidified incubator with 5% CO_2_.

Treatment groups for experiments included high glucose (HG) DMEM and low glucose (LG) DMEM (Sigma-Aldrich), supplemented with either 1 µM bovine serum albumin (BSA) as a control or 1 µM sphingosine-1-phosphate (S1P) as a treatment group to investigate the role of S1P in metabolism. Under normoxic conditions, cells were exposed to atmospheric oxygen conditions at 21% (NO) or 1–2% oxygen for hypoxic conditions (HO).

### 2.2. Mitochondrial Respiration via Oxygen Consumption Rate

Mitochondrial respiration was determined using a Seahorse XFe96 extracellular flux analyzer (Agilent, Santa Clara, CA, USA) as previously described [8]. Single spheroids of a defined size and composition were formed by seeding 2.0 × 10^4^ cells into 96 well round bottom ultra-low adherence plates (Corning, Corning, NY, USA) and centrifuging for 5 min at 900 rpm. All cells were incorporated into small spheroids of approximately the same size by 12 h. To keep treatment times consistent between various times of adherence, the spheroids were transferred to the 96 well cell culture plate (Agilent) used for the assay at the appropriate time before the assay was run 24 h after the initial formation of all spheroids. Adherence time points included 0 h (for which spheroids were transferred directly to assay plates without time for adherence), 4, 8, and 12 h. For some experiments, cells were treated for 24 h and then allowed to adhere for 24 h prior to the assay run.

Preliminary optimization studies to achieve reproducible measurements of the metabolic responses of spheroids in the XFe96 were conducted to determine experimental protocol and inhibitor concentration. Experiments consisted of 3 min mixing, a 2 min wait step, and 3 min measurement cycles. Prior to the assay, the medium was replaced with 180 µL of serum-free, phenol-red-free, bicarbonate-free medium. Oxygen consumption rate (OCR) was measured under basal conditions for three cycles, then in the presence of ATP synthase inhibitor oligomycin (1.0 μmol/L), mitochondrial uncoupler carbonylcyanide-p-trifluoromethoxyphenylhydrazone FCCP (3.0 μmol/L, Sigma), and complex I inhibitor rotenone with cytochrome C oxidase inhibitor antimycin A (1.0 μmol/L rotenone + antimycin A) for five cycles each. All experiments were performed at 37 °C. Total traces as well as basal respiration, maximum respiration (after FCCP addition), spare respiratory capacity (basal OCR—maximum OCR), and ATP synthesis rate (basal OCR—oligo-stimulated OCR) were calculated. Non-mitochondrial respiration, represented by the OCR remaining after the addition of rotenone/antimycin A, was subtracted from all values to report only mitochondrial OCR. Following outgrowth imaging, spheroids were washed 1× with PBS and placed in 30 μL of RIPA buffer for protein analysis by Pierce^TM^ BCA Protein Assay (Thermo Fisher Scientific, Waltham, MA, USA). Data were presented as the mean ± SEM of three independent biological experiments performed in eight or more replicates.

### 2.3. Outgrowth Imaging

Immediately following the completion of the respiration assays, each individual spheroid was imaged using a standard light microscope (Nikon, Minato City, Tokyo, Japan) at 10× magnification to determine adherence outgrowth. Outgrowth and spheroid area were assessed with NIS Elements software and ImageJ. The spheroid area was subtracted from the total outgrowth to determine the measurements of the outgrowth alone. Images were processed using Adobe Photoshop^®^. Data were presented as the mean ± SEM.

### 2.4. Glucose Uptake Imaging

To measure the uptake of glucose in spheroids, cells were seeded at 5 × 10^3^ cells in 96 well ultra-low adherence plates (Corning) and allowed to form for 24 h in the indicated media (HG/LG) and oxygen conditions (NO/HO) at 37 °C. Spheroids were placed onto glas- bottom culture plates (35 mm, Cellvis) with 300 μL of medium and allowed to adhere for 4 or 24 h in the indicated oxygen conditions. At the indicated time points, the medium was removed and spheroids were incubated with 2 μL of fluorescently labeled deoxyglucose (NBD-glucose) (Cayman Chemical, Ann Arbor, MI, USA) in 100 μL PBS. Spheroids were placed back into the appropriate incubator for 15 min, then washed 2× with PBS, and spheroids were imaged in 100 μL PBS using a Leica DMI8 MP confocal microscope at 25× magnification. LASx software was used to create 3D reconstruction and overlay images.

### 2.5. Lactate Secretion

To measure lactate secretion of spheroids, spheroids were generated by seeding 2.5 × 10^5^ cells in 6-well ultra-low adherence plates (Corning) for 24 h in the indicated media (HG/LG) or oxygen (NO/HO) conditions at 37 °C. Spheroids were washed with PBS and transferred to cell culture-treated 6-well plates (Corning) and incubated in 3 mL of phenol red-free, serum-free DMEM (HG/LG) for the indicated time in normoxic or hypoxic conditions. Following adherence time (4 or 24 h), media was diluted five-fold and assayed for lactate concentration using a colorimetric kit according to the manufacturer’s instructions (BRSC, University of Buffalo), and absorption was read by a plate reader (Cytation 5, Biotek, Winooski, VT, USA). Data were normalized by protein content assessed by the Pierce^™^ BCA Protein Assay (Thermo Fisher Scientific). Data were presented as the mean of three independent experiments performed in replicates of three per treatment condition and presented as the mean ± SEM.

### 2.6. Substrate Metabolism

Spheroids were formed by seeding 1.0 × 10^6^ cells per well in 6 well ultra-low adherence plates (Corning), transferred to cell culture-treated 6-well plates (Corning), and allowed to adhere prior to the assay (4 or 24 h). Spheroids were incubated in appropriate media (HG/LG, BSA/S1P) and incubated at 37 °C in normoxic or hypoxic conditions. Fatty acid oxidation in spheroids was measured using radiolabeled [1-^14^C]-palmitic acid (Perkin Elmer, Waltham, MA, USA), measuring ^14^CO_2_ production and ^14^C-labeled acid-soluble metabolites as previously described [40]. Briefly, samples were incubated in 0.5 μCi/mL of [1-^14^C]-palmitic acid for 1 h after which the media was acidified with 200 μL 45% perchloric acid for 1 h to release ^14^CO_2_ which was trapped in a tube containing 1 M NaOH. The vial’s ^14^C concentrations were measured on a 4500 Beckman Coulter scintillation counter (Indianapolis, IN, USA). Acid-soluble metabolites were determined by collecting the acidified media and measuring the ^14^C levels. Glutamine oxidation, TCA flux, and glucose oxidation were measured with methods similar to that of fatty acid oxidation by substitution of [U-^14^C]-glutamine, 2-^14^C pyruvate, and U-^14^C glucose for [1-^14^C]-palmitic acid, respectively. Total protein content for each sample was assessed via bicinchoninic acid assay (Thermo Fisher Scientific) and the data were normalized to total protein and expressed as nm/mg protein/h. Data were presented as the mean ± SEM of three independent biological experiments performed in two replicates per group.

### 2.7. Substrate Utilization

Characterization of mitochondrial utilization of a potential substrate was assessed using Biolog (Hayward, CA, USA) MitoPlate^TM^ S-1 assays as per the manufacturer’s instructions. Spheroids were generated by seeding 1.2 × 10^6^ cells in 6-well ultra-low adherence plates (Corning) in replicates of three 24 h before the assay. Spheroids were transferred to cell culture-treated 6-well plates (Corning) for 4 or 24 h. To form a single cell solution, spheroids were digested in 500 μL of accutase for 2–5 min. DMEM with 4% FBS was added to inactivate the accutase and the three replicates were combined into one tube. The cell solution was put through a 75-micron filter and cells were counted to obtain 1.0 × 10^6^ cells per mL, and the medium was replaced with 1× Biolog MAS. Colorimetric change was observed via kinetic readings on a Cytation 5 (Biotek) at 5% CO_2_ and normoxic (ambient oxygen) or hypoxic (1% oxygen) conditions at 37 °C. Data were presented as the mean ± SEM of the slope of three independent biological experiments performed in three replicates per plate for each condition.

### 2.8. Statistics

Data were presented as the mean ± SEM. For two-component comparisons, data were analyzed using a Student’s two-tailed *t*-test. When comparing spheroids grown in different conditions in comparison to HG NO, a one-way ANOVA with Dunnett’s multiple comparisons test was performed. All results were analyzed in Prism (GraphPad). Results are considered significant at *p* < 0.05. Statistical modeling of OCR data (basal respiration, max respiration, spare respiratory capacity, and ATP synthesis) utilized a linear mixed model analyzed in JMP (SAS Institute) by the Statistical Applications and Innovations Group (SAIG) at Virginia Tech. Linear mixed models are an extension of simple linear models to allow both fixed and random effects. Mixed models allow for more than one source of random variability to account for the correlation of observations within a group or across time, for instance [41,42]. Results were considered significant at *p* < 0.05.

## 3. Results

We have shown that the survival of aggregated ovarian cancer cells is associated with a significantly reduced respiration and highly fragmented mitochondrial phenotype leading to a reduced proliferation [8,31]. This low rate of energy generation may not support the outgrowth of the metastases at secondary sites. Thus, here we investigate the impact of adherence and culture conditions that mimic early time points in vivo metastasis on cellular energetics.

### 3.1. Determination of Spheroid Respiration during Early Adherence

To investigate the effect of adherence on the respiration of MOSE spheroids, we measured the oxygen consumption rate (OCR) over time of adherence using the Seahorse XFe96 to determine when early changes occur that could support the switch from a largely senescent to a proliferative phenotype. In general, the respiration rate was very low, and we did not find significant differences in total OCR traces over time, between culture conditions, or between cell types (Figure 1A,C).

We then calculated basal and maximal respiration, spare respiratory capacity, and ATP synthesis, focusing on the earliest time points of adhesion (4 h) and 24 h. As shown in Figure 1B,D, the calculated parameters were low given the low overall OCR, and data points were variable. At 4 h of adherence, LG NO conditions increased basal respiration (*p* < 0.05) and ATP synthesis (*p* < 0.01) in MOSE-L that was not seen at later time points or in MOSE-L_TIC*v*_ spheroids. Spare respiratory capacity was only increased for LG HO (*p* < 0.05) in MOSE-L at 4 h. At 24 h in MOSE-L, basal respiration was significantly reduced in LG HO (*p* < 0.05), and hypoxia increased spare respiratory capacity (HG HO *p* < 0.001, LG HO *p* < 0.0001). This was also observed for MOSE-L_TIC*v*_ for LG HO (*p* < 0.001); however, this was the only significant finding for the more aggressive cell type.

Treatment of spheroids with the bioactive sphingolipid S1P displayed similar responses to adhesion, glucose, and oxygen levels as the controls (Figure 2). Similar to control spheroids, the OCR remained low for S1P-treated cells without significant differences in the OCR over time, culture condition, or cell type compared to the control (Figure 2A,C, Appendix A).

In MOSE-L, S1P treatment decreased the basal respiration (*p* < 0.05), max respiration (*p* < 0.05), spare respiratory capacity (*p* < 0.05), and ATP synthesis in LG NO at 4 h of adhesion compared to controls (Appendix A). Basal and maximum respiration, spare respiratory capacity, and ATP synthesis calculations were not significantly different in MOSE-L_TIC*v*_ from their controls (Appendix A). After 24 h, only ATP synthesis was higher in HG NO (*p* < 0.05) in MOSE-L (Appendix A), with no differences between control and S1P treatment for MOSE-L_TIC*v*_ (Appendix A). Thus, S1P caused an initial increase in OCR in MOSE-L but this was not maintained over time and the MOSE-L_TIC*v*_ were not responsive to the treatment with S1P compared to the control. These results show that while S1P is elevated in ovarian cancer tissues, plasma [20], and ascites [21], there is no consistent long-term time-, glucose-, or oxygen-dependent increase in the OCR observed in either cell line after S1P treatment in vitro.

#### Statistical Modeling of Variables Contributing to Metabolic Response

Given the nature of comparisons between the independent variables of cell type, time, glucose conditions, oxygen conditions, and treatment with S1P, a linear mixed-effect model was then utilized to determine the relationship of each variable on the dependent variable, OCR, for basal and maximum respiration, spare respiratory capacity, and ATP synthesis. An adjusted natural log transformation was used to address non-normality. The adjustment added a small positive constant to the OCR value in order to ensure valid inputs to the natural log in the case of negative OCR. The fixed effects of cell type (MOSE-L vs. MOSE-L_TIC*v*_), glucose level (HG vs. LG), oxygen level (NO vs. HO), time (as a continuous variable, given as Time and Time^2^), and treatment (control vs. S1P) were compared and also combined to determine the effects on the OCR. In the model, plate to plate variability accounts for a large portion of the overall variability. While the residuals from this model are not normal, mixed model results are robust to violations of assumptions in normality [43]. The model indicated the main variables associated with changes in OCR were cell type, time, cell type x oxygen levels, oxygen treatment x time, and oxygen level x treatment (Table 1). Cell type was associated with increases in basal respiration (*p* < 0.0001) and ATP synthesis (*p* < 0.001) but decreases with spare respiratory capacity (*p* < 0.0001). Time was significantly associated with all calculated parameters (*p* < 0.0001 for all) indicating that over time, OCR increased. It should be noted that the positive linear coefficient of time dominates the overall trend. However, the quadratic time coefficient is negative in some cases which reduces the overall rate of increase from the dominant linear coefficient. Glucose level showed negative associations with maximum respiration and ATP synthesis (both *p* < 0.05), implying an increase in maximal respiration and ATP synthesis with a reduction in glucose level that was not seen with basal respiration or spare respiratory capacity. Oxygen level showed only associations with spare respiratory capacity (*p* < 0.01). Treatment alone did not show associations with any metabolic calculations, but when combined with oxygen level a relationship with basal respiration (*p* < 0.05), maximal respiration (*p* < 0.01), and ATP synthesis (*p* < 0.01) was revealed. This indicates that S1P treatment was not a primary contributing factor to differences in metabolic function as we had hypothesized

### 3.2. Spheroid Outgrowth Increases over Time of Adherence in All Culture Conditions

It is possible that the cells that grow out onto the plastic after adhesion have a different metabolism and thereby affect the measurements above. Thus, spheroids were imaged following mitochondrial assay with the Seahorse XFe96 analyzer (Figure 3A,B) to assess the outgrowth area of spheroids over time of adhesion. The outgrowth area significantly increased in MOSE-L from 4 and 12 h and from 12 to 24 h (Figure 3C). Neither the glucose nor the oxygen levels consistently affected the adhesion and outgrowth of the MOSE-L spheroids when compared to HG NO (Figure 3C). Similar growth patterns were observed in the S1P-treated spheroids across time of adherence and compared to HG NO (Figure 3D). Culturing the cells in LG NO increased the outgrowth over HG NO at 4 h of adhesion in the LG NO group compared to the controls (*p* < 0.01); this was also observed at 24 h (*p* < 0.05). Comparing S1P treatment to control MOSE-L, there were significant increases in outgrowth after S1P compared to the controls (4 h HG HO, *p* < 0.01, LG HO *p* < 0.05; 12 h HG NO *p* < 0.05; 24 h HG NO *p* < 0.05; LG NO *p* < 0.05) (Appendix A) that were unrelated to the growth conditions at the earlier timepoints but restricted to NO after 24 h.

In the highly aggressive MOSE-L_TIC*v*_ spheroids, the outgrowth also significantly increased from 4 to 12 h and between 12 and 24 h which was not affected by glucose or oxygen levels except in LG HO at 4 h (increased, *p* < 0.05). This was also evident in LG NO at 24 h (increased, *p* < 0.05). Further, S1P treatment also caused an overall increase in outgrowth for MOSE-L_TIC*v*_ spheroids (Figure 3D) but compared to the controls only the outgrowth in 4 h LG NO (*p* < 0.05), 12 h HG HO (*p* < 0.05), and 24 h LG HO (*p* < 0.05) were significantly higher in the S1P-treated spheroids without apparent association with culture conditions (Appendix A). When comparing cell types, MOSE-L_TIC*v*_ spheroid outgrowth was initially smaller at 4 h and 12 h, but after 24 h of adhesion, MOSE-L_TIC*v*_ spheroids had significantly larger outgrowth production than MOSE-L spheroids in all conditions (*p* < 0.0001 for all) (Figure 3A–D).

Taken together, outgrowth significantly increased over time of adherence. MOSE-L_TIC*v*_ spheroids typically showed a larger outgrowth area compared to MOSE-L after initially lagging behind in the outgrowth area. Treatment and culture conditions had little effect on outgrowth and while some statistically significant differences were observed, there was no consistent response to glucose, hypoxia, or S1P treatment. Given the major differences in outgrowth between 4 h and 24 h, and the lack of significant differences between other time points during the OCR time trial, the following analyses focused only on investigating 4 and 24 h of adherence to represent the earliest and latest time points of our study.

### 3.3. Time of Adherence Affects Glucose Uptake in Spheroids

We have previously reported that glucose uptake in MOSE spheroids is lower than their adherent counterparts and that it is increased in hypoxia in HG medium in MOSE-L_TIC*v*_ spheroids only, though not to the levels of adherent cells [8]. Given that the spheroid structure with quiescent cells making up the core could mask changes in the uptake in the growing cells on the outer surface and the actively outgrowing cells on the culture dish, we instead sought to visualize the uptake of glucose in spheroids using fluorescently labeled (NBD) glucose. For both MOSE-L and MOSE-L_TIC*v*_ spheroids, glucose uptake was highest in the perimeter cells and adhesion sites while there was very little uptake at the core cells of the spheroids at 4 h (Figure 4) (the top of spheroid larger than 100 mm could not be visualized from the side view). This was not restricted to the cells actively growing on the culture dish (outgrowth) but was also visible in the adhering spheroid cells (shown as the bottom layer of the confocal image stack, 5 µm depth, lower panels of Figure 4A,E). As apparent by the higher intensity of the pixels, culturing the spheroids in LG HO increased glucose uptake at the adhesion sites in both cell lines but not in the spheroid periphery (Figure 4C,G). After 24 h of adherence, the spheroids had flattened and the extended adhesion area and pixel saturation indicate a higher glucose uptake that is still limited to cells at the adhesion sites; no uptake was detected in the top of the spheroids (Figure 4B,F). Culturing the spheroids in LG HO increased glucose uptake in the spheroids (Figure 4D,H) but the increase in the outgrowth area and density of NBD-positive cells at the adhesion sites suggests a higher glucose uptake in MOSE-L_TIC*v*_ grown in LG HO.

MOSE-L_TIC*v*_ spheroids exhibited a higher uptake of glucose than the MOSE-L spheroids at both 4 and 24 h (Figure 4E–H) observed via an increase in signal intensity and area of fluorescence. This confirms our previous results showing adherent MOSE cells increase their glucose uptake during progression [30] and a higher glucose uptake of MOSE-L_TIC*v*_ spheroids in hypoxia [8]. Taken together, our results indicate that adhesion signals increase glucose uptake over time independent of the cells’ phenotype. However, the highly aggressive MOSE-L_TIC*v*_ spheroids display higher amounts of glucose uptake at any culture condition.

### 3.4. Adhesion of Spheroids Increases Lactate Secretion

Lactate secretion is associated with a glycolytic phenotype and is increased during ovarian cancer progression without being affected by aggregation [8,29]. In both MOSE-L and MOSE-L_TIC*v*_ spheroids, lactate secretion significantly increased over time of adherence (Figure 5A,B); this was not different from HG HO or LG NO culture conditions (Figure 5C,D). At both time points, lactate secretion was not significantly higher in the MOSE-L_TIC*v*_ spheroids except at 24 h in HG HO (*p* < 0.01) and LG HO (*p* < 0.05). However, while the culture conditions had no effect on MOSE-L spheroids, LG HO significantly increased the amount of lactate secreted at 24 h compared to HG NO in MOSE-L_TIC*v*_ (*p* < 0.01) (Figure 5A,B). This was not observed in the LG NO due to a significantly higher lactate secretion in HG HO than HG NO, suggesting that the lactate secretion was determined by hypoxia rather than low glucose levels.

### 3.5. Spheroids Switch from Glutamine to Glucose Oxidation over Time of Adherence

Next, we investigated how adhesion affects the oxidation of different substrates over time. Our lab has previously described the preferred glucose over fatty acid oxidation during ovarian cancer progression [30]. Further, glutamine has emerged as a critical player in the anaplerosis of the TCA cycle in cancer cells [44]. Thus, we investigated glucose, fatty acid, and glutamine oxidation over time of adherence. Neither cell line showed significant differences in fatty acid oxidation between culture conditions or across time of adherence (Figure 6A,B, first panels). However, the preference of glucose and glutamine for oxidation switched over time of adherence. For both MOSE-L and MOSE-L_TIC*v*_, glucose oxidation was increased at 24 h (significant for LG HO, *p* < 0.05 and *p* < 0.01 for MOSE-L and MOSE-L_TIC*v*_, respectively) (Figure 6A,B, middle panels). In contrast, glutamine oxidation started higher at 4 h but significantly decreased at 24 h in both culture conditions. Glutamine oxidation trended higher (*p* = 0.09) in MOSE-L_TIC*v*_ cultured in HG NO conditions than in MOSE-L spheroids but was significantly lower when cultured in LG HO (*p* < 0.01), comparable to the levels in MOSE-L under these conditions. Over time, glutamine oxidation was reduced with no significant differences between culture conditions (Figure 6, right panels). To discern if this was switch to glucose oxidation was more affected by glucose or oxygen levels, we also determined substrate utilization on HG HO and LG NO conditions (Appendix A). Substrate oxidation in the MOSE-L_TIC*v*_ appeared to be independent of the glucose concentrations but responsive to hypoxia (increase in fatty acid oxidation, decrease in glucose oxidation at 24 h, decrease in glutamine oxidation at 24 h). These results indicate a switch in the utilization of glucose and glutamine as spheroids adhere, changing from using primarily glutamine to oxidizing more glucose as they adhere. While this increase in glucose oxidation is observed with time, it is also worth noting that glutamine is still oxidized at a significantly higher amount than glucose in both cell types in HG NO and LG HO at 4 h (MOSE-L HG NO (*p* < 0.001); MOSE-L LG HO (*p* < 0.001); MOSE-L_TIC*v*_ HG NO (*p* < 0.001); MOSE-L_TIC*v*_ LG HO (*p* < 0.01) and 24 h (MOSE-L HG NO (*p* < 0.0001); MOSE-L LG HO (*p* < 0.05); MOSE-L_TIC*v*_ HG NO (*p* < 0.0001); MOSE-L_TIC*v*_ LG HO (*p* < 0.01), showcasing the high glutamine utilization of these spheroids. Treatment with S1P did not significantly alter substrate utilization (Appendix A).

### 3.6. Differences in Various Substrate Utilization over Adherence Time

Based on the lack of drastic increase in total respiration over time but the observed switch in substrate oxidation, we next explored which other substrates could be utilized for energy production. Here, we used the Biolog mitochondrial assays which offer 31 substrates from glycolysis and the pentose phosphate pathway, TCA cycle anaplerosis, and fatty acid transport/oxidation. A higher utilization of a substrate is reported through colorimetric change, where a darker color corresponds to a higher degree of electron acceptance via the tetrazolium redox dye. In general, MOSE-L_TIC*v*_ displayed a higher capacity to utilize most substrates compared to the MOSE-L in HG NO conditions with little changes over time (Figure 7A) suggesting that at any point of dormancy or adhesion these highly aggressive cells are not limited to glucose, fatty acids, or glutamine for energy production but can use a variety of substrates if available. MOSE-L cells increased their utilization of several substrates (glucose-6-phosphate, lactic acid, ketobutyric acid, glutamine) at 24 h. Interestingly, MOSE-L spheroids utilized tryptamine in both culture conditions at such a higher degree, especially at 24 h, that we separated the analysis from the other substrates because it masked other changes; MOSE-L_TIC*v*_ did not metabolize tryptamine at all (Figure 7A,B, bottom). In LG HO conditions, the use of almost all substrates was higher in the MOSE-L cells than in HO NO at 4 h but even higher at 24 h of adherence. Glucose, glucose-6-phosphate, gluconate-6-phosphate, and glutamine were the most utilized substrates at 4 h and several substrates showed a robust higher utilization at 24 h, including TCA cycle intermediates (citric acid and isocitric acid) and glycogen in MOSE-L (Figure 7B). Several TCA intermediates were also more utilized in LG HO conditions than in HG NO in the MOSE-L_TIC*v*_ at 24 h (citric, isocitric, and malic acid) and intermediates of glycolysis (glucose-6-phosphate) and the pentose pathway (gluconate-6-phosphate). While these substrates were less utilized after 24 h adhesion in LG HO, glycogen and the glycerol degradation product a-glycerol-PO4 use was higher (Figure 7B). The capacity to utilize other substrates is not the result of the lack of glucose in the LG HO conditions since glucose uptake (see Figure 4) and utilization were even higher under these conditions. Instead, adhesion under hypoxic conditions appears to redirect or reprogram the energy metabolism. These results indicate that both time and culture conditions affect substrate utilization in a cell type-dependent manner and make both cell types more ‘thrifty’.

## 4. Discussion

Metastatic ovarian cancer remains a deadly disease. Aggregation increases the survival of disseminating cancer cells and we have shown previously that the mitochondria in these aggregates are highly fragmented and show a severely suppressed respiration leading to reduced cell growth [8,31]. Here, we investigated how adhesion affects the metabolic phenotype of MOSE cells of different degrees of aggressiveness, mimicking the earliest steps in the metastatic process at distant sites. Since the malignant ascites that transports the ovarian cancer cells throughout the peritoneal cavity is low in oxygen and glucose [17], we combined different glucose and oxygen levels in our analyses of early events in spheroid adhesion. We show here that the respiration of the spheroids was very low in the spheroids and no significant increases in the first 24 h of adherence were noted. Adhesion was associated with a higher lactate secretion and glucose uptake, especially in hypoxia, and a change from utilization of mostly glutamine as the primary energy source to increasing glucose use while maintaining a high glutamine use for TCA cycle anaplerosis. Both cell types were able to utilize other substrates for energy generation in a culture condition-, time-, and phenotype-specific manner. Thus, our studies show that adhesion causes changes in cellular energy metabolism and substrate utilization. Importantly, the most aggressive phenotype of the MOSE-L_TIC*v*_ cells was characterized by a resilience to culture conditions, i.e., low glucose or oxygen levels had less of an effect on their metabolic response, and the capacity to utilize most substrates that are available for energy production at a higher rate than the MOSE-L representing slow-developing disease.

It is important to note that within the experimental data for spheroids there was a high degree of variability in the OCR. While experimental methods and times were kept consistent between Seahorse runs, e.g., defined spheroid size and culture conditions, the consistency of wells that worked within the machinery changed often and within one group OCR could range from 1 to >100 pmol/min/mg protein. A contributing factor to this variability in addition to the low rate of respiration of the spheroids is presumably the location of the spheroid within each well (center vs. rim of the well) which differed despite efforts to plate spheroid consistently, given their small size and influence of fluid and landing pattern as the spheroid was plated. These drastic variations in OCR could explain the lack of significant changes in respiration of both the controls and S1P-treated cells, reducing our ability to see causative effects of adhesion and treatment. Further, the glucose uptake was only apparent in the adherent cells, both from the bottom of the spheroids and the outgrowth. The bulk of the spheroid was unresponsive, indicating that only a small sub-population altered their metabolism upon adherence, confirming a strong adhesion signal we have previously observed that also increased the appearance of polarized mitochondria at adhesion sites (unpublished data). This implies that only cells that are in contact with the culture dish actually have the capacity (increased polarized mitochondria) and glucose availability (increased glucose uptake) to increase OCR and that the metabolically unresponsive bulk of the spheroid may mask changes that occur over time. Despite these limitations and variations, the analysis of the time trials in the linear mixed model suggests an association of time and cell type with the level of OCR during adhesion. Other methodologies to specifically measure oxygen consumption and ATP generation may be more useful in small cancer spheroids with a defined size and low respiration.

S1P is elevated in the malignant ascites and has been shown to promote mTOR signaling, which may stimulate fatty acid synthesis and uptake [45]. In relation to mitochondrial function, S1P has been associated with increases in mitochondrial biogenesis in hepatocarcinoma [46] and is involved with mitochondrial dynamics and homeostasis [47]. In studies of cardiomyocytes and HeLa cells, S1P and sphingosine kinase 2 (an isoform found on the mitochondria) were found to increase oxidative phosphorylation and respiratory chain organization [48]. Given the interactions of S1P with migration, proliferation, and invasion in cancer [23], as well as the interactions between mitochondrial S1P and the assembly of the respiratory chain and oxidative phosphorylation [48], we expected that S1P treatment would cause an increase in cellular metabolism. However, the effects of S1P treatment on metabolic outputs was contrary to our original hypothesis, and there were no consistent effects of S1P on calculations of basal, max respiration, spare respiratory capacity, ATP synthesis, and outgrowth in any condition across time points, culture conditions, and cell types. Both control and treatment groups saw similar metabolic outputs during the mitochondrial stress tests, as well as similar measured outgrowth. If this is due to the concentrations used, or the need for other factors found in the ascites to be effective, or if S1P has targets other than mitochondrial respiration and metabolism needs to be investigated in more detail.

Elevated glucose uptake and lactate secretion in cancer cells independent of oxygen levels are the hallmarks of the Warburg effect [27,28]. Previous investigations in our lab have shown a reduction in glucose uptake after aggregation that was less severe in MOSE-L_TIC*v*_ in hypoxic conditions; lactate secretion was little affected by aggregation [8]. Here, we show that not the availability of glucose in the medium per se but the adhesion to the culture dish provides a strong signal for glucose uptake since no NBD-glucose was observed in the bulk of the spheroid even after 24 h of adhesion independent of the glucose concentration in the medium. Uptake was higher in the MOSE-L_TIC*v*_ and in hypoxic conditions; since these spheroids are larger than those of the MOSE-L cells, the more extensive adhesion area where the glucose uptake is observed may generate a stronger adhesion signal that may be the determining factor. While this is a novel observation, studies in non-adherent colon cancer spheroids also show a glucose uptake only in the oxygenated outer regions of non-adherent spheroids but not the hypoxic core; however, this was dependent on the glucose concentration [49] and spheroid size [50,51]. Thus, it is possible that the exposure time determined by the high fluorescence levels at adhesion sites missed the uptake of glucose in the cells at the upper periphery of the spheroids.

Adhesion increased lactate secretion with the highest levels seen in MOSE-L_TIC*v*_ in hypoxic conditions. Lactate can increase angiogenesis via the stimulation of VEGF expression in endothelial cells, as well as cell migration, wound healing, and repair. Exposure to lactate increases expression of the lactate transporter MCT1 in other cell types, coordinating metabolism that benefits the cancer cells [28]. Lactate can also help support the self-sufficiency of cancer cells, being produced by more glycolytic cells and taken up by more oxidative cells to support oxidative phosphorylation and metabolic symbiosis between cells [52]. This process is dynamic; as spheroids adhere to secondary sites, releasing more lactate to promote vascularization, metabolic symbiosis, and acidification of the microenvironment could help support successful outgrowth.

Cancer cells have demonstrated the ability to use a wide variety of fuel sources for energy besides glucose and lactate. One substrate that has emerged as a metabolic target due to its utilization in energy production and anaplerosis is glutamine [53], the most abundant amino acid in the body. Cancer is often described as being “addicted” to glutamine, given that through glutaminolysis and conversion into α-ketoglutarate it anaplerotically replenishes the TCA cycle for intermediates and electron carriers for energy [53,54,55] or can be used as a nitrogen and carbon source for the synthesis of amino acids, lipids, and nucleotides [45,56]. By replenishing the TCA intermediates with glutamine, intermediates such as citrate or malate can fuel other biosynthetic processes such as lipid synthesis [17,57,58]. In the present study, we observed a switch in substrate oxidation between glutamine and glucose, two critical nutrients tied to cancer metabolism [59]. Oxidation of glucose increased while glutamine oxidation decreased in both cell types over time, suggesting a metabolic switch activated by adhesion. While the glucose levels had no effect, hypoxia increased glucose utilization in both cell lines but more so in the MOSE-L_TIC*v*_ spheroids. This parallels the higher glucose uptake upon adhesion and suggests that in addition to providing energy, glucose provides glycolysis intermediates that can be used for nucleotide, lipid, and protein synthesis in order to generate macromolecules for growth and division [60] seen in the cells growing out from the spheroid. However, both glucose and glutamine are used to fulfill requirements for energy at the early stages of spheroid adhesion. While adherent MOSE cells showed a preference for glucose over fatty acid oxidation during progression [30], here, we did not detect changes in fatty acid oxidation in response to culture conditions or over time of adherence.

Using the Mitoplates, we investigated additional potential energy substrates the cells could use if available and found that the spheroids could use a large number of different substrates that were affected by the time of adherence, indicating that adhesion signals are effectively re-programming cellular metabolism. In general, the aggressive MOSE-L_TIC*v*_ were able to use more substrates for energy production and achieved higher energy yield than the MOSE-L spheroids. This includes glutamine, glucose and its metabolites, fatty acid metabolites, and TCA cycle intermediates. While mutational changes in isocitric dehydrogenase lead to the generation of oncometabolites that can drive carcinogenesis [61], there is little information available on the effect of hypoxia on the activity of TCA enzymes in cancer cells. However, the use of citric and isocitric acid and other TCA intermediates suggest that the enzymes transforming these substrates and generating electrons are fully functional in the MOSE cells. Interestingly, glycogen was highly used by MOSE-L_TIC*v*_, especially in hypoxia. Glycogen is accumulated in ovarian and other cancers [62] and the high expression of its degradation enzyme glycogen, phosphorylase B, in ovarian cancer contributes to proliferation, migration, and invasion in vitro, and tumor development in vivo [63]. Further, glycogenolysis results in enhanced glycolysis, elevated levels of glucose-6-phosphate, and anaplerosis of the TCA cycle [64], providing the opportunity to use TCA intermediates for macromolecule biosynthesis and proliferation. Another substrate of note that was observed to be highly utilized by the MOSE-L but not the MOSE-L_TIC*v*_ spheroids was tryptamine, a metabolite of tryptophan. We observed high utilization of this substrate in MOSE-L, especially at 24 h, regardless of culture condition. Tryptamine is a metabolite of tryptophan, a result of tryptophan decarboxylation. Its function as a neurotransmitter, vasoconstrictor vasodilator, and neuromodulator has been described as has the inhibition of gluconeogenesis by tryptamine [65]; its use as an energy substrate, however, is still unknown. However, as the metabolic plates provide levels of substrates to the cells that do not limit enzymatic reactions, they provide insights into potential substrate use based on availability; to determine if this occurs in physiological conditions, a comprehensive metabolomic analysis is needed.

## 5. Summary and Conclusions

Together, to our knowledge, this is the first study to identify a metabolic switch in oxidation substrates in the earliest stages of adhesion of ovarian cancer spheroids that could support successful outgrowth and metastasis. The respiration of the spheroids was very low and highly variable; albeit limited by the variability of data between assay plates, time and cell type were associated with increases in the OCR, suggesting adhesion signals may impact cellular respiration. Upon adherence over time, ovarian spheroids display an increase in glucose uptake and lactate secretion, a decrease in glutamine oxidation, and an increase in glucose oxidation. The aggressive phenotype of the MOSE-L_TIC*v*_ spheroids was characterized by a significantly larger outgrowth after 24 h of adherence, higher glucose uptake and lactate secretion, a more drastic switch from glucose to glutamine utilization over time, and the capacity to utilize more substrates and to a greater extent than the MOSE-L spheroids. Further, MOSE-L_TIC*v*_ spheroids were less affected by low glucose conditions. Our results indicate that the flexibility in substrate utilization in response to the dynamic changes in ascites composition that alters substrate availability and the reprogramming of metabolism in response to adhesion signals may be critical for the survival of disseminating ovarian aggregates and the support of secondary outgrowth of ovarian metastases. These results support our hypothesis that the aggregation-induced suppression of metabolism is reversible in the initial steps of spheroid adhesion and contributes to successful disease progression. This identifies the adhesion-induced metabolic reprogramming as a druggable target for the prevention of metastatic outgrowth in women with ovarian cancer and may suppress or delay metastases at distant sites and prolong the life of women with ovarian cancer. Future studies will investigate the molecular mechanisms of this switch as it occurs over time.

## Figures and Tables

**Figure 1 cancers-14-01399-f001:**
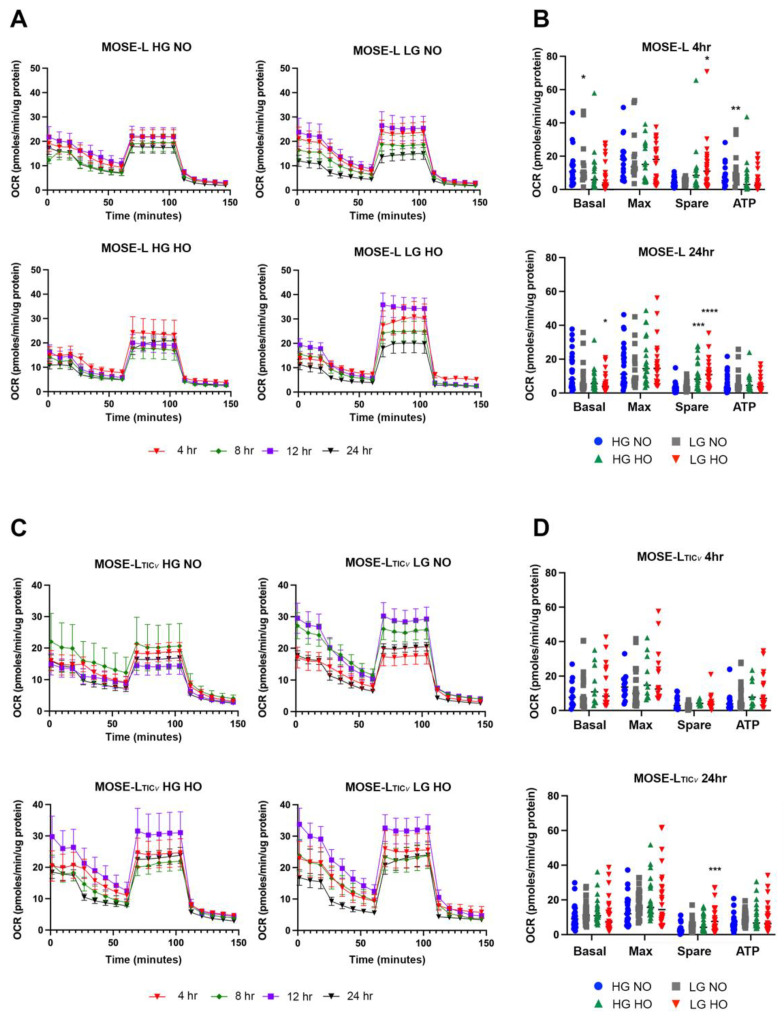
Oxygen consumption rate (OCR) and metabolic outputs in spheroids over time. OCRs were graphed as time traces to compare overall data (**A**) MOSE-L and (**C**) MOSE-L_TIC*v*_. Metabolic outputs were calculated from OCR trace values for (**B**) MOSE-L and (**D**) MOSE-L_TIC*v*_. Data are represented as the mean ± SEM and ANOVA with Dunnett’s multiple comparisons test to compare to HG NO. * *p* < 0.05 compared to HG NO; ** *p* < 0.01 compared to HG NO; *** *p* < 0.001 compared to HG NO; **** *p* < 0.0001 compared to HG NO. Abbreviations: high glucose (HG); low glucose (LG); normoxia (NO); hypoxia (HO).

**Figure 2 cancers-14-01399-f002:**
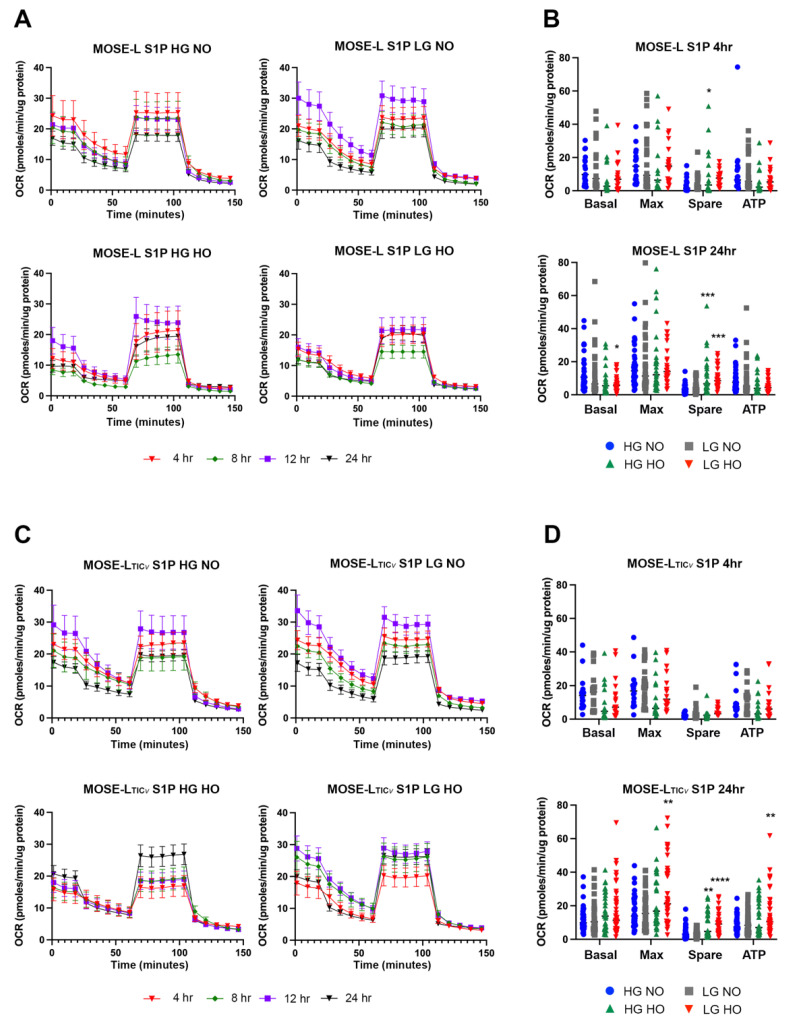
Oxygen consumption rate (OCR) and metabolic outputs in S1P-treated spheroids. (**A**) MOSE-L time traces, (**C**) MOSE-L_TIC*v*_ time traces. Metabolic outputs were calculated from OCR trace values for (**B**) MOSE-L and (**D**) MOSE-L_TIC*v*_ treated with S1P. Data are represented as the mean ± SEM and ANOVA with Dunnett’s multiple comparisons test to compare to HG NO. * *p* < 0.05 compared to HG NO; ** *p* < 0.01 compared to HG NO; *** *p* < 0.001 compared to HG NO; **** *p* < 0.0001 compared to HG NO. Abbreviations: high glucose (HG); low glucose (LG); normoxia (NO); hypoxia (HO).

**Figure 3 cancers-14-01399-f003:**
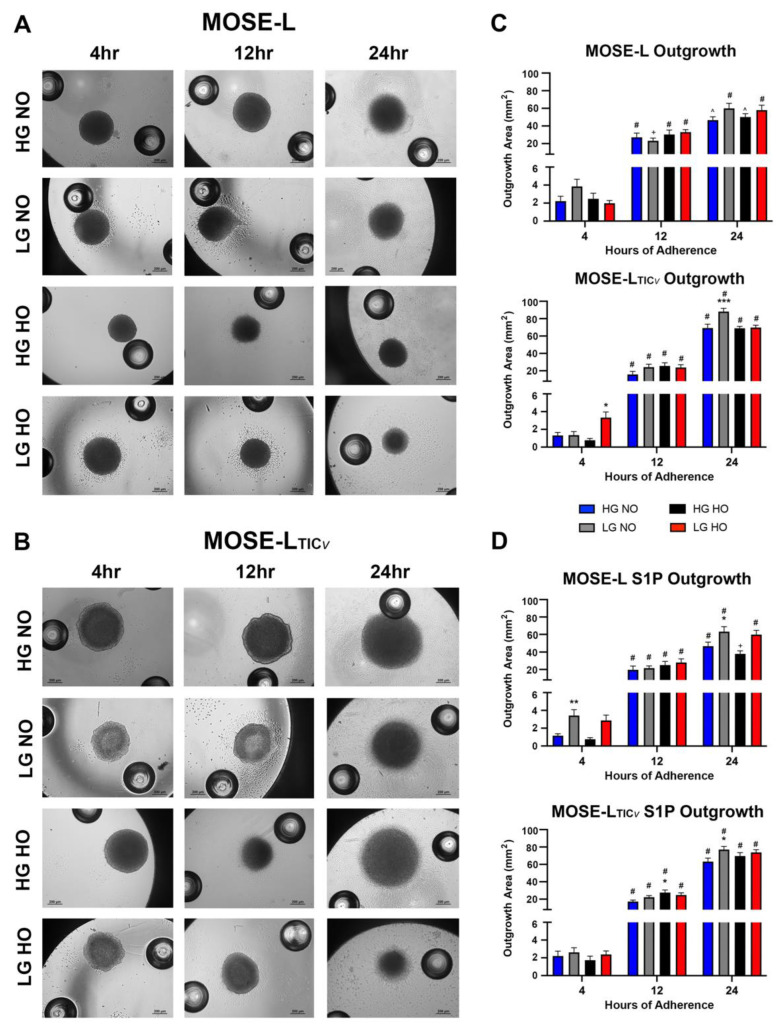
Spheroid outgrowth increased over time of adherence. Outgrowth of spheroids was measured in MOSE-L and MOSE-L_TICv_ in brightfield images quantified using NIS Elements and ImageJ (**A**) Representative images of MOSE-L spheroids (**B**) Representative images of MOSE-L_TIC*v*_ spheroids, (**C**) Quantification of outgrowth for MOSE-L and MOSE-L_TICv_ spheroids, and (**D**) for S1P-treated MOSE-L and MOSE-L_TICv_. Data are represented as the mean ± SEM. * *p* < 0.05 compared to HG NO; ** *p* < 0.01 compared to HG NO; *** *p* < 0.001 compared to HG NO. + *p* < 0.05 compared to previous time point; ^ *p* < 0.01 compared to previous time point; # *p* < 0.001 compared to previous time point. Abbreviations: high glucose (HG); low glucose (LG); normoxia (NO); hypoxia (HO).

**Figure 4 cancers-14-01399-f004:**
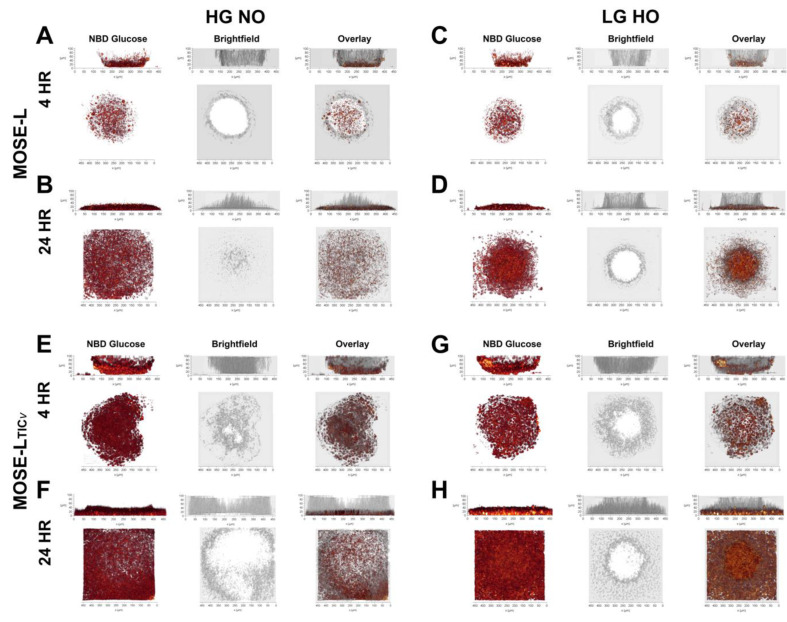
Glucose uptake in spheroids at 4 and 24 h in different conditions. Images represent 3D reconstructions of confocal imaging and reconstruction of NBD-glucose, brightfield, and overlay. Measurements were taken in MOSE-L and MOSE-L_TICv_ spheroids in HG NO and LG HO at 4 and 24 h of adherence. Spheroids are displayed as viewed from the side (top image) and from the top (bottom image). (**A**) MOSE-L 4 h HG NO (**B**) MOSE-L 24 h HG NO (**C**) MOSE-L 4 h LG HO (**D**) MOSE-L 24 h LG HO (**E**) MOSE-L_TIC*v*_ 4 h HG NO (**F**) MOSE-L_TIC*v*_ 24 h HG NO (**G**) MOSE-L_TIC*v*_ 4 h LG HO (**H**) MOSE-L_TIC*v*_ 24 h LG HO. Abbreviations: high glucose (HG); low glucose (LG); normoxia (NO); hypoxia (HO).

**Figure 5 cancers-14-01399-f005:**
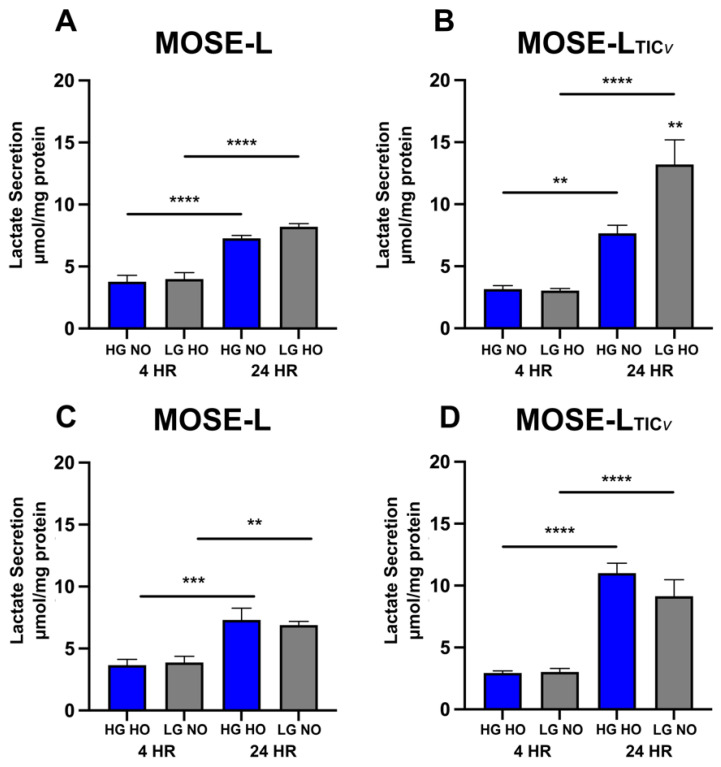
Lactate secretion of spheroids at 4 and 24 h of adhesion. Lactate secretion was measured in spheroids adherent for 4 or 24 h in optimal (HG NO) and limiting (LG HO) conditions in (**A**) MOSE-L and (**B**) MOSE-L_TIC*v*_ as well as HG HO and LG NO in (**C**) MOSE-L and (**D**) MOSE-L_TIC*v*_ spheroids. Data are represented as the mean ± SEM. ** *p* < 0.01 compared to HG NO; bars with ** *p* < 0.01 compared to 4 h; bars with *** *p* < 0.001 compared to 4 h; bars with **** *p* < 0.0001 compared to 4 h. Abbreviations: high glucose (HG); low glucose (LG); normoxia (NO); hypoxia (HO).

**Figure 6 cancers-14-01399-f006:**
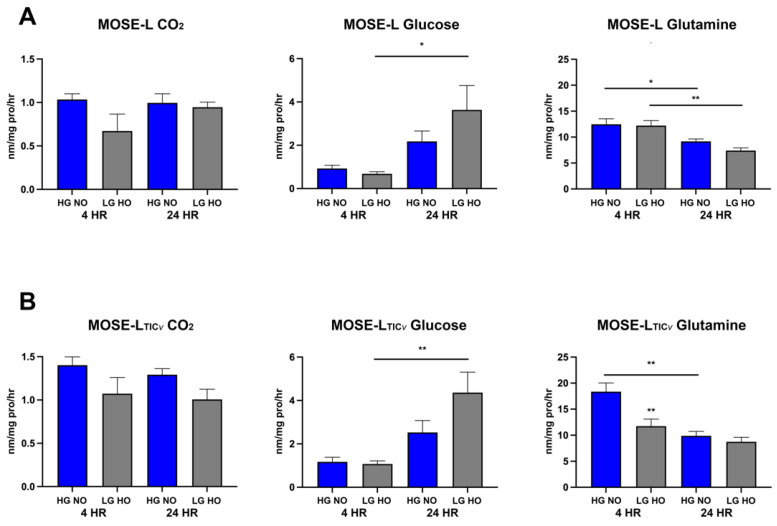
Substrate oxidation of ^14^C-palmitic acid into CO_2_, ^14^C-glucose, and ^14^C-glutamine in HG NO and LG HO culture conditions in spheroids. (**A**) MOSE-L substrate oxidation (**B**) MOSE-L_TIC*v*_ substrate oxidation at 4 and 24 h of adherence. Data are represented as the mean ± SEM. ** *p* < 0.01 compared to HG NO within the same time point; bars with * *p* < 0.05 compared to 24 h; bars with ** *p* < 0.01 compared to 24 h. Abbreviations: high glucose (HG); low glucose (LG); normoxia (NO); hypoxia (HO).

**Figure 7 cancers-14-01399-f007:**
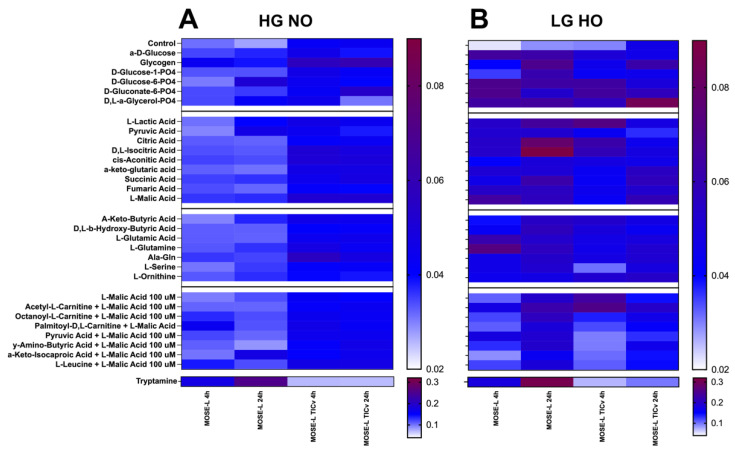
Substrate utilization using MitoPlates™ heat map. Differential utilization of substrates for energy production from intermediates for glycolysis, the TCA cycle, and fatty acid oxidation shown through colorimetric redox assay for MOSE-L and MOSE-L_TICv_ at 4 and 24 h of adherence in (**A**) HG NO and (**B**) LG HO. Tryptamine heat map is shown at the bottom of (**A**,**B**) with its own heat map quantification scale. Abbreviations: high glucose (HG); low glucose (LG); normoxia (NO); hypoxia (HO).

**Table 1 cancers-14-01399-t001:** Summary table of the fixed-effect models. * *p* < 0.05, ** *p* < 0.01, *** *p* < 0.001, **** *p* < 0.0001. Abbreviations: high glucose (HG); hypoxia (HO); bovine serum albumin (BSA).

Variables	Basal Resp.	Max Resp.	Spare Resp. Cap.	ATP Synthesis
Cell Type [MOSE-L_TIC*v*_]	0.184 ****	ns	−0.071 ****	0.111 ***
Glucose Level [HG]	ns	−0.194 *	ns	−0.168 *
Oxygen Level [HO]	ns	ns	0.125 **	ns
Time	0.026 ****	0.024 ****	0.008 ****	0.018 ****
Time^2^	−0.003 **	−0.002 *	ns	−0.003 **
Treatment [BSA]	ns	ns	ns	ns
Cell Type × Glucose Level	ns	−0.043 *	ns	−0.039 *
Cell Type × Oxygen Level	0.099 ****	ns	−0.026 **	0.059 ***
Cell Type × Time	ns	ns	0.003 *	0.008 **
Cell Type × Treatment	ns	ns	ns	ns
Glucose Level × Oxygen Level	ns	ns	ns	ns
Glucose Level × Time	ns	ns	ns	ns
Glucose Level × Treatment	ns	ns	ns	ns
Oxygen Level × Time	0.016 **	0.013 **	0.006 **	0.011 **
Oxygen Level × Treatment	0.060 *	0.064 **	ns	0.058 **
Time × Treatment	ns	ns	ns	ns
Time^2^ × Cell type	ns	ns	ns	ns
Time^2^ × Glucose Level	ns	ns	ns	ns
Time^2^ × Oxygen Level	ns	ns	ns	ns
Time^2^ × Treatment	ns	ns	ns	ns

## Data Availability

The data presented in this study are available in this article.

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
