# Peer review of "Metabolic Reprogramming of Ovarian Cancer Spheroids during Adhesion"

_cancers, 2022, doi:10.3390/cancers14061399_

Round 1

Reviewer 1 Report

The authors hypothesize that the adherence of spheroids leads to a metabolic switch. The manuscript reports changes during adhesion and outgrowth of ovarian cancer spheroids in two previously described mouse cell lines with different aggressiveness. They use a variety of methods to demonstrate these changes. The manuscript is nicely written, with several minor issues that need to be checked.

Lines 271-272: you claim LG HO conditions increase basal respiration and ATP synthesis in MOSE-L at 4h – I don not see this on fig 1B and the stars are not placed there on the figure. Instead, they are placed on the LG NO groups.

Lines 273-276: this sentence in not clear to me. It looks like spare respiratory capacity in increased after 4 h in LG HO conditions, but after 24h spare respiratory capacity was increased for both HG HO and LG HO (hypoxic conditions). It took me several read through the sentence and careful examinuation of figure 1B and D to figure out what you were trying to say here. It should also be mentioned here that you do not see most of these effect in the more aggresive variant of your cells (except for the spare respiration increase in LG HO after 24h).

Line 342-343: There are no stars on the panel C that show this. Please add the stars.

Wherever you compare the data between S1P treated and untreated cells, this is impossible to see on the graphs that are presented in the paper. I suggest adding the comparison of S1P-treated and untreated measurements as a separate figure (maybe as a supplementary figure), so it is easier to follow/compare. This specifically refers to descriptions from lines 281-286, lines 344-348 and lines 3761-363.

Figure 3: you show the outgrowth of spheroids beautifully, but some spheroids are very different in size (e.g. MOSE-LTICv LG NO 12h vs. 24h), almost like they were made under different magnification. Please check if this are the correct images to use.

Line 385 – please check the sentence, extra words here

Line 424 – „At both time points, lactate secretion was significantly higher in the MOSE-LTICv spheroids...“. – not really. It may be true for 24h, but not 4h according to the data shown here.

Line 434 – this should be LG NO?

Author Response

please see attached letter

Reviewer 2 Report

The authors have previously shown that survival of aggregated ovarian cancer cells is associated with significantly reduced respiration and highly fragmented mitochondrial phenotype. The authors claim that the low proliferation rate seen under these conditions is associated with low rate of energy generation that may not support outgrowth of the metastases at secondary sites.  For this reason, the authors endeavored to investigate the impact of adherence and culture conditions that mimic early time points in vivo metastasis on cellular energetics.

Overall. the way the data presented throughout the manuscript is very confusing and not clearly stated leading the reader to struggle to interpret the outcome of these studies. One of the major concerns is all of the studies were conducted with MOSE cells in vitro.  Why the authors did not perform similar studies in high grade serous cell lines, (assuming that the authors did not have access to patient derived ascites cells) is not clear. This lack of human derived cells is a major drawback of this study. The outcome form using human OC cell lines or more importantly, patient derived ascites cells may show a different profile.  The authors should address this weakness in their discussion. Language and grammatical errors need to be fixed. On a positive note, the quality of the figures throughout the manuscript was excellent.

  1. The authors state that in general they did not find significant differences in total OCR traces over time, between culture conditions, or between cell types with respiration rate present at low levels. However, the authors discussed that LG HO conditions increased basal respiration (p<0.05) and ATP synthesis (p<0.01) in MOSE-L at 4hrs, (Figure 1B) and the data shown in the figure doesn’t show any change or p value for LG HO group at 4hr. The authors should provide an explanation for their statement.
  2. Figure 2B shows that in MOSE-L, S1P treatment increased the basal respiration (p<0.05), max respiration (p<0.05), and spare respiratory capacity (p<0.05) in LG NO at 4 h of adhesion, but the figure doesn’t show any p value significance for this.
  3. The authors claimed that S1P caused an initial increase of OCR in MOSE-L but this was not maintained over time and the MOSE-LTICv were not responsive to the treatment with S1P. However, the figure 2D suggest that there is a significant increase in basal, maximum and ATP synthesis in MOSE-LTICv LG HO group at 24hrs upon S1P treatment. Additionally, their statement that “there is no consistent long-term time-, glucose-, or oxygen-dependent increase of the OCR observed in either cell line after S1P treatment in vitro” is inconsistent with the data presented.
  4. Overall, the presentation of results for figures 1 and 2 is too confusing to follow and should be written with proper details and in a more simplified manner without overinterpreting the data.
  5. For figure 3C MOSE-L outgrowth no p-value was shown in the figure as described in result section. The convoluted presentation of the results took away the impact of the study.
  6. For figure 3C MOSE- LTICv graph shows significant increase at 4hr for LG HO and at 24hrs for LG NO. The authors have failed to explain the impact of these changes.
  7. Quantification of glucose uptake assay should be provided for figure 4.
  8. Figure 5 lacks a 0hr control time point. Without 0hr timepoint, how can the authors claim there is an increase in lactate secretion at 4hr?
  9. Figure 6 also lacks a 0hr control.
  10. No survival assays were performed to understand the effect of the conditions/nutrients for the normal vs TIC cells.
  11. While the discussion was comprehensive, it reads more like a review. The authors should clearly provide a concise concluding summary statement highlighting the impact of the outcome from these studies in terms of prognosis of patients with this deadly disease including future implication of this study.
  12. Please define all the abbreviations used for a particular figure in the figure legends (like HG, LG, NO, HO...)

Author Response

please see attached letter

Round 2

Reviewer 2 Report

The authors have addressed the concerns